# The Role of MicroRNAs in Dilated Cardiomyopathy: New Insights for an Old Entity

**DOI:** 10.3390/ijms232113573

**Published:** 2022-11-05

**Authors:** Elena Alonso-Villa, Fernando Bonet, Francisco Hernandez-Torres, Óscar Campuzano, Georgia Sarquella-Brugada, Maribel Quezada-Feijoo, Mónica Ramos, Alipio Mangas, Rocío Toro

**Affiliations:** 1Research Unit, Biomedical Research and Innovation Institute of Cadiz (INiBICA), Puerta del Mar University Hospital, 11009 Cádiz, Spain; 2Medicine Department, School of Medicine, University of Cadiz, 11002 Cádiz, Spain; 3Medina Foundation, Technology Park of Health Sciences, 18016 Granada, Spain; 4Department of Biochemistry and Molecular Biology III and Immunology, Faculty of Medicine, University of Granada, 18016 Granada, Spain; 5Cardiology Service, Hospital Josep Trueta, University of Girona, 17007 Girona, Spain; 6Cardiovascular Genetics Center, Institut d’Investigació Biomèdica de Girona (IdIBGi), 17190 Salt, Spain; 7Centro de Investigación Biomédica en Red, Enfermedades Cardiovasculares (CIBERCV), 28029 Madrid, Spain; 8Medical Science Department, School of Medicine, University of Girona, 17003 Girona, Spain; 9Arrhythmias Unit, Hospital Sant Joan de Déu, University of Barcelona, 08950 Barcelona, Spain; 10Cardiology Department, Hospital Central de la Cruz Roja, 28003 Madrid, Spain; 11Medicine School, Alfonso X el Sabio University, 28007 Madrid, Spain; 12Internal Medicine Department, Puerta del Mar University Hospital, School of Medicine, University of Cadiz, 11009 Cadiz, Spain

**Keywords:** dilated cardiomyopathy, microRNA, inflammation, reticulum endoplasmic stress, oxidative stress, apoptosis, autophagia, fibrosis, molecular pathways, etiology

## Abstract

Dilated cardiomyopathy (DCM) is a clinical diagnosis characterized by left ventricular or biventricular dilation and systolic dysfunction. In most cases, DCM is progressive, leading to heart failure (HF) and death. This cardiomyopathy has been considered a common and final phenotype of several entities. DCM occurs when cellular pathways fail to maintain the pumping function. The etiology of this disease encompasses several factors, such as ischemia, infection, autoimmunity, drugs or genetic susceptibility. Although the prognosis has improved in the last few years due to red flag clinical follow-up, early familial diagnosis and ongoing optimization of treatment, due to its heterogeneity, there are no targeted therapies available for DCM based on each etiology. Therefore, a better understanding of the mechanisms underlying the pathophysiology of DCM will provide novel therapeutic strategies against this cardiac disease and their different triggers. MicroRNAs (miRNAs) are a group of small noncoding RNAs that play key roles in post-transcriptional gene silencing by targeting mRNAs for translational repression or, to a lesser extent, degradation. ﻿A growing number of studies have demonstrated critical functions of miRNAs in cardiovascular diseases (CVDs), including DCM, by regulating mechanisms that contribute to the progression of the disease. Herein, we summarize the role of miRNAs in inflammation, endoplasmic reticulum (ER) stress, oxidative stress, mitochondrial dysfunction, autophagy, cardiomyocyte apoptosis and fibrosis, exclusively in the context of DCM.

## 1. Introduction

Dilated cardiomyopathy (DCM) is a heart muscle disease characterized by left ventricle (LV) or biventricular dilatation and systolic dysfunction in the absence of coronary artery disease, hypertension, valvular disease or congenital heart disease [1]. DCM is one of the most common causes of heart failure (HF), with an estimated prevalence of approximately 1:250–400 and up to 1:2500 in the general population [2].

Despite decades of research on DCM, there is no consensus with respect to an unique classification of the disease, as this entity is an “umbrella” that includes many processes that lead to a common end [1]. Traditionally, the main etiologies are ischemic (ICM) and non-ischemic DCM, whose diagnosis is based on the presence of severe atherosclerotic lesions in the coronary arteries [3]. Non-ischemic DCM includes several causes, such as infection (mostly viral), inflammation, autoimmune diseases and exposure to drugs, toxins or chronic alcohol abuse, among others. Idiopathic DCM (iDCM) is assigned after the identification of systolic dysfunction and morphologically enlarged left ventricles, when an etiological assessment is undertaken and no identifiable cause is found [2]. It has been estimated that iDCM may constitute more than 70% of all diagnoses of non-ischemic DCM. Genetic factors are an important cause of iDCM, and familial transmission has been reported in 20–35% of cases [4]. Familial DCM is mostly transmitted in an autosomal dominant inheritance pattern. During the past two decades, familial genetic studies have associated approximately 40 genes with this disease, mostly encoding structural and sarcomeric myocardial proteins. The most prevalent are titin (*TTN*) and lamin A/C (*LMNA*) genes, as they are responsible for nearly 10–15% and up to 35% of genetic DCM [5]. Other well-known causal genes of DCM are those that encode cardiac troponin T2 (*TNNT2*), phospholamban (*PLN*), desmin (*DES*), tropomyosin (*TPM1*), vinculin (*VCL*), and RNA-binding motif protein 20 (*RBM20*) [6,7].

Several mechanisms such as inflammation, endoplasmic reticulum (ER) stress, mitochondrial dysfunction, oxidative stress, autophagy, apoptosis and fibrosis have been described to play a critical role in the progression of LV dysfunction, adverse LV remodeling and, ultimately, in the development of HF associated with DCM (reviewed in Section 3). Therefore, it may be useful to fully understand the signaling pathways that control these mechanisms and their impact on each DCM etiology to develop novel therapeutic strategies against DCM.

MicroRNAs (miRNAs) are highly conserved, small, single-stranded non-coding RNAs with an average length of 18–22 nucleotides, which negatively regulate gene expression at the post-transcriptional level. The mechanisms of action are based on the blockage of protein translation and/or mRNA degradation by binding to complementary sequences of the 3′-untranslated region (3′-UTR) of mRNAs [8]. In general, there is only partial complementarity between the nucleotides of the miRNA seed region, a sequence of six to eight nucleotides at the 5′ end, and the target mRNA region. Thus, perfect complementarity between miRNAs and mRNAs is not essential to reduce gene expression and, in fact, rarely occurs in mammals. Moreover, a single mRNA can contain multiple binding sites for various miRNAs, thus generating a complex network of miRNA–mRNA interactions [8]. MiRNAs regulate approximately 30% of all human protein-coding genes and play an important role in the regulation of biological processes by establishing intricate networks that involve multiple miRNA–mRNA interactions [9].

Over 2500 miRNAs have been identified in humans [10], and one third of the human genome is estimated to be regulated by miRNAs [11]. However, the function of many of them remains unknown. MiRNA deregulation has been associated with almost all diseases in humans, including cardiovascular diseases (CVDs) [12]. In this regard, miRNAs have been identified as critical regulators in the initiation and maintenance of left ventricular hypertrophy, ischemic heart disease, HF, hypertension and arrhythmias [12]. In pathological conditions, their expression levels can be found to be deregulated in circulation and/or in cardiac tissue [13]. In the last decade, many studies have reported a link between the miRNA expression profile and DCM, and many of them have been identified as potential biomarkers for diagnosis and prognosis of DCM, as well as to discriminate between etiologies [14,15,16]. In addition, several studies have elucidated a key role of miRNAs in regulating the distinct mechanisms that underlie DCM progression. In this review, we will summarize the current knowledge of the role of miRNAs in different mechanisms involved in DCM, such as inflammation, ER stress, oxidative stress, mitochondrial dysfunction, autophagy, cardiomyocyte apoptosis and fibrosis. Moreover, we discuss how the knowledge gained on the role of miRNAs that control these mechanisms can be used to generate new therapeutic approaches for the treatment of DCM.

## 2. Inflammation and miRNAs

As in most CVDs, the role of inflammation in DCM is a natural response to injury and is an important mechanism for healing and tissue repair. However, an inadequate or persistent inflammatory state increases tissue damage and poor healing. Accumulating data have revealed a relevant inflammatory component in the pathogenesis of DCM, as immune cell activation is often present in DCM pathological examinations [17]. Cytokines have been found to play an important role in the pathogenesis and pathophysiology of DCM, promoting fibrosis and cardiac remodeling [18]. It is well-established that pro-inflammatory cytokines, such as tumor necrosis factor alpha (TNF-α), interleukin-6 (IL-6), interleukin-1β (IL-1β) and interleukin-4 (IL-4), are produced by different etiologies of DCM, including iDCM, ICM and familial DCM [17,19,20].

MiRNAs have emerged as key regulators of inflammation, as they modulate signaling of the onset and termination of inflammation; depending upon the target mRNA, miRNAs may either promote or suppress inflammation [21,22,23]. Recent studies have identified distinct miRNAs as regulators of inflammation during DCM (Figure 1 and Table 1). Corsten and colleagues reported that the *miR-221/-222 cluster* orchestrates the antiviral and inflammatory immune response to viral myocarditis, which is closely related with DCM [24]. *MiR-221/-222* was found to be upregulated in cardiac tissue of mice during myocarditis caused by coxsackievirus B3. In addition, miR-221 and miR-222 were upregulated by coxsackievirus B3 infection in neonatal rat cardiomyocytes. First, it was found that *miR-221/-222* had antiviral properties, as *miR-221/-222* inhibition stimulated Coxsackievirus B3 replication in isolated cardiomyocytes. Secondly, *miR-221/-222* also regulated the immune response, as its systemic inhibition increased the influx of T-cells and macrophages upon infection. In situ hybridizations of miR-221 in normal mouse hearts revealed expression by multiple cell types in the heart, including cardiomyocytes, endothelial cells, and interstitial immune-like cells. In mouse hearts with induced viral myocarditis, this expression was strongly apparent in the zones of inflammation. Similar results were found in cardiac biopsies from viral myocarditis patients. Finally, this study showed that miR-221 and miR-222 directly target the pro-inflammatory transcription factors E26 transformation-specific sequence 1 and 2 (*ETS1*, *ETS2)* (Figure 2) [24]. More recently, miR-223-3p was found to be downregulated in the plasma of a mouse model of autoimmune myocarditis. The transfer of miR-223-3p-overexpressing dendritic cells attenuated the inflammatory response in these mice. In addition, dual-luciferase reporter assay demonstrated that miR-223-3p directly targets the *Nlrp3* inflammasome (Figure 2). Further in vitro assays showed that miR-223-3p regulated dendritic cell function via the NLR family pyrin domain-containing 3 (*Nlpr3*) inflammasome [25]. Finally, LaRocca et al. showed that transgenic mice with cardiomyocyte-specific miR-152 overexpression developed DCM. Genome-wide transcriptional profiling of these hearts and subsequent Gene Ontology biological processes enrichment analysis demonstrated that the upregulated genes were mainly involved in the inflammatory response [26]. In rats, Sheu et al. showed that the implantation of induced pluripotent stem cell-derived mesenchymal stem cells that overexpressed miR-19a-3p and miR-20a-5p into the LV of a rat model of DCM induced by doxorubicin improved inflammation and left ventricular ejection fraction [27]. Finally, an in vitro experiment demonstrated that miR-146a, upregulated in cardiac fibroblasts (CFs) infected with coxsackievirus B3, downregulated proinflammatory cytokine expression, disrupting the nuclear factor kappa-light-chain-enhancer of activated B cells (NF-κB) pathway through targeting the toll-like receptor 3 (*TLR3*) and the TNF receptor-associated factor 6 (*TRAF6*) (Figure 2) [28].

In humans, miR-155 and miR-133a were found to be up-regulated in endomyocardial biopsies from patients with inflammatory DCM, as compared to non-inflammatory DCM. Moreover, their expression levels were correlated with inflammatory cell count in the inflammatory DCM cohort [29]. In this regard, microarray analysis revealed a distinct pattern of miRNA expression in CD4+ T cells from DCM patients as compared to the controls, suggesting a role of miRNAs in the aberrant activation of CD4+ T cells. Among them, miR-451a, downregulated in DCM CD4^+^ T cells, was identified as a regulator of cell activation and proliferation [30]. Beg at al. found that exosomal miR-146a was increased in the inflammatory response of HF patients with systolic dysfunction. Concordantly, miR-146a expression increased in the cardiomyocyte cell line H9c2 upon tumor necrosis factor alpha (TNF-α) treatment [31]. Finally, Yang et al. reported that circulating levels of miR-181b were diminished in patients with HF with reduced ejection fractions, compared with healthy individuals. Moreover, miR-181b expression was suppressed before increasing TNF-α, IL-1β and IL-6 levels in the heart tissue of a rat model of HF induced by isoproterenol. miR-181b overexpression decreased TNF-α, IL-1β and IL-6 expression in a primary culture of rat neonatal cardiomyocytes [32].

## 3. ER Stress and miRNAs

Over the last decade, ER stress has been shown to be involved in DCM pathogenesis, among other CVDs [33,34]. The ER is responsible for the synthesis, folding, and quality control of most proteins in the cell. Perturbations of this homoeostasis lead to an accumulation of misfolded and unfolded proteins, a process known as ER stress [35]. ER stress induces a process termed the unfolded protein response (UPR), which corrects the protein-folding defect and reestablishes ER homoeostasis. However, if the UPR fails to restore this balance, the UPR assumes an adverse role and triggers apoptosis [33,34]. The UPR is initiated by the activation of three signaling branches, namely inositol-requiring enzyme-1α (IRE1α), protein kinase R (PKR)-like endoplasmic reticulum kinase (PERK) and activating transcription factor 6 (ATF6). In normal conditions, IRE1α, ATF6 and PERK are bound to the ER chaperone, immunoglobulin-binding protein (Bip)/glucose-regulated protein 78 (GRP78), to maintain its active state [36]. In stressed conditions, GRP78 dissociates from IRE1α, PERK, and ATF6, leading to UPR activation [37,38].

Numerous studies have reported a link between miRNAs and ER stress in the heart, as miRNAs and members of the ER stress pathway can modulate each other in CVD. In 2012, Belmont and colleagues showed that, besides its role in regulating the transcription of potentially protective genes, ATF6 also regulates miRNA expression in mouse hearts [39]. This work was the first study to report that UPR members modulate miRNA expression in the heart. On the other hand, several miRNAs have been shown to regulate transcription factors of the UPR in the heart. In 2013, Zhang et al. reported that miR-702 regulates the expression of *Atf6* in heart tissue in an isoproterenol-induced ER stress mouse model. Luciferase reporter assays confirmed that miR-702 directly targets *Atf6* [40]. As is the case for miR-702, other miRNAs have been reported to regulate ER stress through directly targeting members of the UPR in the heart. MiR-199-5p regulates UPR activation in cardiomyocytes by targeting *ATF6* and *GRP78* [41], whereas miR-1a-3p has been shown to inhibit *GRP94* by binding to the 3′UTR of this gene [42].

However, little is known about the regulatory role of miRNA modulating ER stress during DCM. MiR-194-5p inhibition has been shown to attenuate ER stress-related doxorubicin cardiotoxicity in H9c2 cells via P21-activated kinase 2 (PAK2) and XBP1s both in vivo and in vitro [43]. PAK2 has been reported to play a cardioprotective role by improving ER function through the IRE1/XBP1 pathway [44]. Luciferase assays demonstrated that *Pak2* is a direct target of miR-194-5p (Figure 2) [43]. More recently, two complementary studies performed by our group found that miR-16-5p was overexpressed in plasma from ICM patients compared to healthy controls. MiR-16-5p was identified as a crucial regulator of the ATF6-mediated cytoprotective response upon ER stress activation in AC16 human cardiomyocyte-like cells (Figure 2). This work demonstrated that *ATF6* is a direct target of miR-16-5p and its suppression reverted tunicamycin-induced *ATF6* downregulation in AC16 cells [45,46] (Figure 1,Table 1).

## 4. Mitochondria Dysfunction and miRNAs

The large quantities of energy needed for the heart metabolism links mitochondrial impairments to several processes in DCM [47,48]. In addition, mitochondria are the central hub of cellular metabolism, providing metabolites for biosynthesis and are an important source of reactive oxygen species (ROS). On the other hand, communication between mitochondria and other organelles, especially the ER, is mandatory for cardiomyocyte homeostasis [47].

Various miRNAs have been described to contribute to distinct CVDs, such as HF, ischemic heart diseases, cardiac hypertrophy and diabetic cardiomyopathy, through the regulation of relevant signaling pathways and mitochondrial function-related proteins [49]. In this regard, several studies that used transgenic mice with overexpressed specific miRNA activity have demonstrated a crucial role in the initiation and progression of DCM through regulating genes involved in the maintenance of mitochondrial function (Figure 1 and Table 1). The complexes of the electron transport chain and ATP synthase comprise the oxidative phosphorylation (OXPHOS) system, which is essential to provide the energy for cardiac function [50]. Mutations in mitochondrial-encoded OXPHOS genes result in the development of DCM [50,51,52]. Wijnen and colleagues reported that a transgenic mouse line that specifically overexpressed miR-30c in cardiomyocytes developed severe DCM [53]. Microarray-based gene expression profiling of miR-30c transgenic hearts before onset of the DCM phenotype indicated mitochondrial dysfunction. This was confirmed by the downregulation of the OXPHOS complexes III and IV at the protein level, indicating an impaired oxidative capacity of the cardiomyocytes. Moreover, the observation of decreased HSP60 proteins, a chaperone involved in mitochondrial maintenance, indicated mitochondrial homeostasis impairment [53]. Similarly, transgenic mice with cardiomyocyte-specific miR-7 overexpression developed DCM, which was accompanied by reduced levels of NDUFA9, a key component of NADH dehydrogenase complexes. Oxygen consumption analysis of purified mitochondria from isolated cardiomyocytes from miR-7 transgenic hearts showed significant reductions in the maximum respiration rate compared to controls, therefore confirming that overexpression of miR-7 in cardiomyocytes leads to mitochondrial dysfunction. In addition, transmission electron microscopy showed alteration in mitochondrial ultra-structures [54]. MiR-1a-3p also regulated mitochondrial oxidative phosphorylation by increasing the expression of NADH dehydrogenase subunit 1 (*ND1*) and cytochrome c oxidase subunit 1 (*COX1*), which are involved in the coding of complex I and complex IV of the electron transport chain. The intravenous injection of agomiR-1 increased the left ventricular ejection fraction and the expression of *Nd1* and *Cox1* in a mouse model of isoproterenol-induced HF [55].

It has been shown that obesity affects the production of non-cardiac factors, such as adipokines, and miRNAs, which may have an impact on cardiac function through modulating mitochondrial function [56,57]. In this context, several studies have reported a critical role of circulating miRNAs in regulating cardiac function and mitochondrial activity in obesity-related cardiomyopathy. In particular, increased levels of circulating miR-194 were correlated with impaired cardiac function and mitochondrial activity in this disease [58]. Exosomes from obese mice increased miR-194 expression in mouse primary cardiomyocytes. In addition, functional analysis of mitochondria from these cultured cardiomyocytes showed decreased basal oxygen consumption and electron transport chain complex I activity [58]. miR-194 sponges also increased ATP production, basal oxygen consumption and complex I activity in cardiac tissue from high-fat diet-fed mice [58]. Similar findings were also reported for exosomal miR-29a and miR-122 [59,60]. However, in contrast to miR-194 and miR-29a, whose specific regulatory mechanisms are not clear, miR-122 impaired mitochondrial function through direct binding to ADP-ribosylation factor-like 2 (*ARL2*) (Figure 2) [59].

Based on the ER–mitochondria crosstalk [61,62], the previously mentioned study by Toro et al. demonstrated that miR-16-5p-mediated ER stress promoted mitochondrial dysfunction in human cardiac cells [46].

## 5. Oxidative Stress and miRNAs

Oxidative stress is defined as the excess production of ROS relative to antioxidant defense [63]. Under physiological conditions, cardiac ROS signaling plays a fundamental function in cell homeostasis; however, excess ROS production has been associated with the pathophysiology of DCM and HF [46,64].

In the heart, ROS are primarily produced by the mitochondria, NADPH oxidases (NOX), xanthine oxidase, and uncoupled nitric oxide synthase [64]. Multiple antioxidant defense systems exist to block harmful effects of ROS by converting ROS into non-toxic molecules. These can be divided into enzymatic, which includes catalase, glutathione peroxidase, superoxide dismutase, and glutaredoxins; and non-enzymatic, which includes vitamins E and C, beta-carotene, ubiquinone, lipoic acid, urate, and reduced glutathione [65].

Whereas the crosstalk between miRNAs and ROS in cardiac diseases, such as cardiac hypertrophy, HF, myocardial infarction, or ischemia/reperfusion injury, has been extensively studied [66,67], little is known about the miRNA–ROS crosstalk in DCM (Figure 1 and Table 1). MiR-448-3p has been reported to play a critical role in the development of ROS-induced DCM [68]. Several studies have demonstrated that NADPH oxidase 2 (*NOX2*) expression, one of the major NOX isoforms expressed in the heart, and NOX2-derived ROS production are involved in the pathophysiology of Duchenne muscular dystrophy, a muscle-wasting disease accompanied by DCM [69,70]. Kyrychenko et al. observed drastic downregulation of miR-448-3p in dilated hearts from a mouse model of Duchenne muscular dystrophy (*mdx*). In vivo experiments using *mdx* mice demonstrated that miR-448-3p inhibition increased the expression of the neutrophil cytosolic factor 1 (*Ncf1*) gene, which encodes the NOX2 regulatory subunit p47phox. Luciferase assays confirmed *Ncf1* as a direct target of miR-448-3p (Figure 2). Accordingly, systemic inhibition of miR-448-3p in wild-type mice increased *Ncf1* expression, NOX2-dependent ROS production, and induced DCM [70]. On the other hand, miR-132 expression was found to be downregulated in peripheral blood of patients with HF associated with DCM. MiR-132 overexpression protected H9c2 cells against oxidative stress by increasing their antioxidant capacity [71]. More recently, Chiang et al. found miR-144 to be downregulated in heart tissue from a mouse model of Friedreich’s ataxia, a disease generally associated with severe DCM. The downregulation correlated with overexpression of the antioxidant transcription factor nuclear factor erythroid 2-related factor 2 (*Nrf2*), which has been previously reported as a direct target of miR-144 (Figure 2) [72,73]. Finally, Ogawa et al. discovered that transgenic mice with cardiomyocyte-specific miR-143 overexpression develop a DCM-like phenotype. Although the expression of hexokinase 2 (*Hk2*), a cardioprotective gene that is a target of miR-143 (Figure 2), was strongly suppressed in the heart of these mice, no significant difference in HK2 activity between transgenic and non-transgenic mice was observed in vitro. Surprisingly, glutathione (GSH) production and the GSH-to-oxidized glutathione ratio were increased in the hearts of 3-month-old transgenic mice, indicating that miR-143 overexpression leads to a reductive, rather than an oxidative, redox state [74].

Recent findings suggest a crosstalk between oxidative stress and ER stress [75,76,77]. In the context of DCM, Toro et al. provided evidence that miR-16-5p may constitute a key player, mediating the crosstalk between ATF6-mediated ER stress and ROS production in the heart [46]. As mentioned above, this study demonstrated enriched expression of miR-16-5p in plasma from ICM patients, which was accompanied by an increase in the oxidative stress markers *malondialdehyde* and advanced oxidation protein products. In addition, miR-16-5p overexpression in human cardiomyocytes also resulted in increased advanced oxidation protein product levels [46].

## 6. Autophagy and miRNAs

Autophagy has emerged as a key player in DCM pathology [78,79,80,81,82,83]. Autophagy is a conserved process involved in maintaining cellular homeostasis, eliminating dysfunctional long-lived cytosolic proteins and organelles in a lysosomal-dependent way. The autophagy process is essential in preserving cardiac homeostasis in physiological settings [84]. Mounting evidence has revealed that miRNAs play an important role in the regulation of autophagy during cardiac disorders, including myocardial infarction, cardiac hypertrophy, cardiac fibrosis, and HF [85]. However, its role in regulating autophagy during DCM has been less explored (Figure 1 and Table 1).

The mammalian mechanistic target of rapamycin (mTOR) is a crucial regulator of autophagy induction. mTOR inhibits autophagy induction by negatively phosphorylating the unc-51-like kinase (ULK) complex, which is composed of ULK1/2, autophagy-related gene 13 (*ATG13*), autophagy-related 101 (*ATG101*), and focal adhesion kinase family interacting protein with a mass of 200 kDa (*FIP200*) [86]. Transgenic mice with cardiac-specific overexpression of miR-222 developed a DCM phenotype. Furthermore, these transgenic mice displayed sign of autophagy inhibition, as protein levels of the autophagy markers microtubule-associated protein 1 light chain 3 alpha (LC3) and sequestosome 1 (P62) were deregulated in heart tissue. In addition, mTOR phosphorylation was significantly increased, indicating activation of the mTOR pathway [87]. MiR-144 was found to regulate its target gene *mTOR* in a mouse model of ICM (Figure 2) [88]. The complete loss of miR-144 in mice subjected to permanent ligation of the left anterior descending coronary artery resulted in a worse HF phenotype, with marked systolic dysfunction and LV dilatation. The Western blot analysis showed upregulation of phosphor-mTOR and total mTOR in miR-144 knockout hearts after surgery [88]. Accordingly, P62 levels were also increased in miR-144 knockout mice, as compared to wild-type ones [88,89]. Caragnano et al. reported that miR-22, a stronger inhibitor of autophagy, was highly enriched in ventricular biopsies of transplanted DCM patients. After immunohistochemistry, cellular and molecular biology and metabolomics analysis in hearts explanted from 62 DCM patients, these authors reported misfolded protein accumulation, aggresome formation, and significantly increased levels of mTOR phosphorylation, indicating mTOR activation and autophagy inhibition. Consistently, a reduction in the nuclear localization of the EB transcription factor, a master regulator of lysosomal biogenesis, was observed in DCM hearts [90]. Finally, miR-16-5p, an miRNA enriched in plasma from ICM patients, was proposed to promote autophagy in human cardiomyocytes by targeting *ATG14*, an essential autophagy-specific regulator of the class III PI3K complex (Figure 2). In addition, miR-16-5p overexpression in AC16 cells downregulated *ATG14* levels and resulted in autophagosome accumulation [45].

Upon autophagy induction, the PI3K complex, which is composed of PI3KC3/hVPS34, p150 and Beclin-1 (BECN1), is activated, triggering the formation of the autophagosomes. In normal conditions, the Bcl2 family anti-apoptotic proteins interact with Beclin-1 and inhibit autophagy [91,92]. However, when autophagy is activated, Beclin-1 is liberated from BCL2 and induces autophagosome formation. In an in vitro model of doxorubicin-induced cardiotoxicity in rat primary cardiomyocytes, miR-30e was shown to suppress autophagy by directly targeting *Becn1* in an in vitro model of doxorubicin-induced cardiotoxicity in rat primary cardiomyocytes (Figure 2) [93], which shows a similar phenotype to DCM [94].

## 7. Cardiomyocyte Apoptosis and miRNAs

Apoptosis is a form of programmed death [95]. Patients with DCM show cardiomyocyte apoptosis and increased intercellular space filled with granulation tissue composed of macrophages, endothelial cells and fibroblasts [96]. Hence, inhibiting the progression of cardiomyocyte apoptosis is one of the most important targets of DCM treatment. The apoptotic process is triggered by the intrinsic mitochondrial pathway and the extrinsic cell surface death receptor pathway. This process is characterized by a series of structural and morphological changes, including chromatin condensation, DNA degradation, cell shrinkage, and blebbing of plasma membranes [97]. The major apoptotic pathway is initiated by the release of cytochrome *C* from mitochondria in response to an apoptotic stimulus. Released cytochrome *C*, in the presence of dATP, forms an activation complex with apoptotic protein-activating factor-1 (APAF1) and caspase-9 (CASP9) that activates downstream caspases (cysteine-aspartic proteases, cysteine aspartates or cysteine-dependent aspartate-directed proteases) [98,99,100], playing a key role in programmed cell death. This complex promotes a group of antiapoptotic proteins, such as BCL2, and proapoptotic proteins, such as BAX, resulting in the final morphological and biochemical alterations [101,102]. Moreover, the balance between BCL2/BAX may be important in the increased rate of apoptosis in cardiac myocytes [103,104].

Whereas some miRNAs, such as miR-122, miR-208a, miR-137-3p, miR-1 and miR-9, have been described to promote cell apoptosis in the pathogenesis of myocardial disorders, others, such as let-7a-5p, miR-133a, miR-613, miR-24, miR-135a, miR-210, miR-26a-5p, miR-369, miR-129-5p, miR-147 and miR-184, have been identified to enhance cardiomyocyte survival, inhibiting apoptosis in myocardial infarction [105].

In recent years, multiple studies have demonstrated that miRNAs have an instrumental role in apoptosis and in the pathogenesis of DCM (Figure 1 and Table 1). Microarray analysis of ventricle tissue from a validated murine phospholamban mutant model of DCM revealed 24 differentially expressed MRNAs [106]. In vitro assays of neonatal cardiomyocytes showed that, among these miRNAs, individual suppression of miR-1, miR-29c, miR-30c, miR-30d, miR-149, miR-486 and miR-499 induced apoptosis [106]. Computationally integrated transcriptomic, proteomic and functional annotation data predicted the calcium-/calmodulin-dependent protein kinase 1D *Camk1d*, which is linked to apoptosis in murine erythroleukemia [107], the Wnt pathway antagonist with multiple connections to apoptosis *Apc* (adenomatous polyposis coli) [108], and the ubiquitin-conjugating enzyme *Ube2d3*, which supports NRDP1-mediated degradation BRUCE, triggering apoptosis [109], as putative target genes of miR-29c, mir-499 and miR-30d, respectively [106]. In humans, miR-150 was found to be downregulated in plasma of ICM patients [110]. In vivo and in vitro experiments revealed miR-150 as a stress-responsive protector against cardiomyocyte apoptosis, directly inhibiting the pro-apoptotic early growth response 2 gene (*EGR2*) (Figure 2) [111]. Additionally, miR-675 has been demonstrated to promote cardiomyocyte apoptosis, targeting the apoptotic suppressor gene proliferation-associated 2G4 (*Pa2g4*) (Figure 2) [112]. The expression of miR-675 was significantly upregulated in the myocardium of a DCM-rat model induced by Adriamycin. Moreover, both mRNA and protein levels of PA2G4 were reduced in DCM rats. Luciferase reporter assays demonstrated that *Pa2g4* is a direct target of miR-675 [112]. Li et al. showed that the long noncoding RNA LINC00339 and miR-484 constitute an axis that regulates doxorubicin-induced cardiomyocyte apoptosis. LINC00339 expression was increased in both primary cardiomyocytes and H9c2 cells exposed to doxorubicin. Knockdown of LINC00339 inhibited cardiomyocyte apoptosis. Interestingly, an opposite expression trend was detected for miR-484. Luciferase reporter assays indicated that LINC00339 directly targets miR-484 (Figure 2). Accordingly, gain- and loss-of-function assays showed a reciprocal regulation between LINC00339 and miR-484. Thus, knockdown of LINC00339 resulted in the upregulation of miR-484 and decreased apoptosis, whereas when co-transfected with the miR-484 inhibitor, this resulted in significantly increased apoptosis in primary cardiomyocytes and H9c2 [113].

Cardiomyocyte apoptosis plays a crucial role in alcoholic DCM. MiR-186-5p expression was found to be upregulated in AC16 cells upon ethanol treatment, increasing the apoptotic process; the luciferase reporter assays demonstrated that miR-186-5p directly targets the anti-apoptotic gene (X-linked inhibitor of apoptosis) *XIAP* (Figure 2) [114]. More recently, the plasmacytoma variant translocation 1 (PVT1)/miR-23a-3p/caspase-10 (CASP10) axis was reported to induce cardiomyocyte apoptosis in an in vitro model of high glucose-induced DCM. In this study, miR-23a-3p was found to be downregulated, whereas *PVT1* and *CASP10* were upregulated in AC16 cells upon high glucose treatment. Accordingly, miR-23a-3p overexpression or knock down of *PVT1* were able to reverse high-glucose induced apoptosis. Luciferase reporter assays confirmed the interaction of *PVT1* and *CASP10* with miR-23a-3p (Figure 2). Finally, this study proposes that PVT1 might increase *CASP10* expression via sponging miR-23a-3p, thus inducing apoptosis [115].

## 8. Cardiac Fibrosis and miRNAs

Cardiac fibrosis, characterized by accumulation of collagen following cardiomyocyte death, occurs early in the progression of DCM, increasing cardiac rigidity, decreasing myocardial performance, and enhancing the risk of HF, sudden cardiac death and malignant arrhythmias [116,117]. Since the adult heart is not able to regenerate its muscular tissue, cardiomyocytes are substituted by CFs, which proliferate and differentiate into myofibroblasts, a more contractile and highly proliferative cell phenotype. Once induced, myofibroblasts produce and secrete greater levels of extracellular matrix components, such as collagen fibers, fibronectin, and profibrotic mediators. Macrophages, monocytes, T lymphocytes, mast cells, and endothelial cells are also involved in this process [118,119,120,121]. Several molecular pathways and humoral factors, including transforming growth factor beta (TGFβ), the renin-angiotensin-aldosterone system, TNF-α, IL-1β, and IL-6, have been reported to play a key role in modulating the profibrotic process [122,123,124,125,126].

Diverse studies have shown that miRNAs play a central role in regulating the pathogenesis of myocardial remodeling by modulating cardiac fibrosis, angiogenesis, and inflammatory response through multiple mechanisms [127]. In the context of DCM, a regulatory role of miRNAs in fibrosis has been also elucidated (Figure 1 and Table 1). MiR-208 is a cardiac-specific miRNA encoded within intron 27 of the isoforms of myosin heavy chains α-MHC and is essential for the expression of the genes involved in cardiac contractility and fibrosis [128]. Satoh et al. found significantly higher expression levels of miR-208 in endomyocardial biopsies of patients with DCM that were correlated with the myocardial collagen volume [129]. MiR-699a was first shown to be downregulated in cardiac progenitors from *Sgcb-null* hearts, a mouse model of limb–girdle muscular dystrophy type 2E affected by DCM with extensive regions of fibrosis, and fatty infiltrations [130,131]. In addition, Quattrocelli et al. demonstrated that intraventricular delivery of adeno-associated viral vectors induced long-term miR-669a overexpression and improved survival of *Sgcb-null* mice. Treated hearts displayed a significant decrease in hypertrophic remodeling and fibrosis [132]. Bernardo et al. showed that miR-34a is upregulated in hearts from a mouse model of DCM with cardiac-specific overexpression of mammalian sterile 20-like kinase 1 (*Mst1*) [133,134]. Moreover, in vivo inhibition of miR-34a decreased the expression of collagen 3 (*Col3a1*) [133]. In healthy hearts, high levels of miR-26, miR-29, miR-30 and miR-133a and low levels of miR-21 are required to balance extracellular matrix turnover by keeping TGFβ and connective tissue growth factor (CTGF) proteins at low levels. In this sense, it was observed that the expression of miR-26, miR-29, miR-30 or miR-133a decreases, while miR-21 expression increases in animal models during pathological remodeling, which may relieve the expression of pro-fibrotic genes, resulting in enhanced collagen synthesis and fibrosis [135]. Accordingly, the expression of these ﻿fibrosis-linked miRNAs both in plasma and myocardial biopsies has been correlated with fibrosis in DCM patients [136,137]. Nishiga et al. reported that miR-33 knockout mouse hearts show reduced fibrotic responses to pressure overload and diminished CF proliferation. Moreover, the expression of miR-33a in endomyocardial biopsies of DCM patients was correlated with improving hemodynamic parameters [138]. Significantly lower levels of miR-221/-222 were found in myocardial biopsies from patients with DCM or aortic stenosis with severe fibrosis, as compared to the matched patients with non-severe fibrosis. Interestingly, a negative correlation was observed between miR-221/222 levels and the extent of myocardial fibrosis in aortic stenosis patients. Inhibition of both miRNAs during angiotensin II-mediated pressure overload in mice resulted in increased fibrosis and aggravated LV dilation and dysfunction. Furthermore, miR-221/222 inhibition stimulated TGFβ-mediated profibrotic SMAD2 signaling in rat CFs, whereas overexpression of both miRNAs had the opposite effect. Finally, the miRNA-221/222 family may target several genes involved in TGFβ signaling, including *JNK1* (c-Jun N-terminal kinase 1), the receptors *TGFβR1* and *TGFβR2*, and *ETS-1* (ETS proto-oncogene 1) [139]. It has been shown that suppression of androgen receptors reduces cardiac inflammation and fibrosis in a mouse model of autoimmune inflammatory DCM [140]. In addition, androgen receptor administration was shown to enhance the expression of miR-125b [141], an miRNA upregulated in fibrotic human hearts [142]. Wang et al. confirmed this concept using an in vitro assay of mouse primary CFs. Thus, the administration of a degradation enhancer of androgen receptors in primary CFs attenuated the expression of miR-125b, which subsequently inhibited the generation of collagen, whereas their overexpression had the opposite effect. Moreover, miR-125b inhibition after androgen receptor overexpression attenuated TGFβ-induced collagen I and α-SMA expression, and reduced CF proliferation [141]. More recently, an integrated analysis of hub genes and miRNAs using data of DCM patients from the GEO database (GSE112556) identified miR-144-3p and miR-9-3p as potential regulators of myocardial fibrosis in DCM that target the fibronectin 1 (FN1) gene, an extracellular matrix component [143]. Finally, a prospective phase 1a study of intracoronary cardiosphere-derived cell infusion in children with DCM and reduced ejection fractions demonstrated that cardiosphere-derived cell treatment improved cardiac function and myocardial fibrosis. MiRNA analysis on the small RNA fraction of cardiosphere-derived cells and their secreted exosomes, isolated from individuals and compared with that of human CFs, showed that the expression of exosomal miR-146a-5p is correlated with the reduction in both myocardial fibrosis and extracellular volume fraction [144].

The main miRNAs involved in the regulation of cardiac fibrosis in the pathogenesis of DCM are reported in Table 1.

**Table 1 ijms-23-13573-t001:** Summary of reported miRNAs in the distinct mechanisms involved in DCM progression. miRNA: microRNA; nRCMs, neonatal rat cardiomyocytes; CFs, cardiac fibroblasts; PC, primary cardiomyocyte. Note: Asterisk means not experimentally validated.

Mechanisms	miRNA	Species	Tissue/Cell	Target	Expression	References
Inflammation	miR-223-3p	Mouse	Plasma	*Nlpr3*	Downregulated	[25]
	miR-152	Mouse	Heart	*-*	Upregulated	[26]
	miR-221/-222	Mouse, rat	Heart, nRCMs	*ETS*, *ETS2*	Upregulated	[24]
	miR-19a-3p, miR-20a-5p	Rat	Plasma	*-*	Downregulated	[27]
	miR-146a	Mouse, Human	CFs, exosomes	*TLR3*, *TRAF6*	Upregulated	[28,31]
	miR-155, miR-133a	Human	Heart	-	Upregulated	[29]
	miR-451a	Human	CD4^+^ T cells	-	Downregulated	[30]
	miR-181b	Human	Plasma	-	Downregulated	[32]
ER stress	miR-194-5p	Rat	H9c2	*Pak2*	Upregulated	[43]
	miR-16-5p	Human	Plasma	*ATF6*	Upregulated	[46]
Mitochondrial dysfunction	miR-30c	Mouse	Heart	-	Upregulated	[53]
	miR-7	Mouse	Heart	-	Upregulated	[54]
	miR-1a-3p	Mouse	Heart	-	Downregulated	[55]
	miR-194	Human	Plasma	-	Upregulated	[58]
	miR-29a	Human	Plasma	-	Upregulated	[60]
	miR-122	Human	Plasma	*ARL2*	Upregulated	[59]
Oxidative stress	miR-448-3p	Mouse	Heart	*Ncf1*	Downregulated	[68]
	miR-144	Mouse	Heart	*Nrf2*	Upregulated	[72]
	miR-143	Mouse	Heart	*Hk2*	Upregulated	[74]
	miR-132	Human	Plasma	-	Downregulated	[71]
	miR-16-5p	Human	Plasma	*ATF6*	Upregulated	[46]
Autophagy	miR-222	Mouse	Heart	-	Upregulated	[87]
	miR-144	Mouse	Heart	*mTOR*	Downregulated	[88]
	miR-30e	Rat	Heart, PC	*Becn1*	Downregulated	[93]
	miR-22	Human	Heart	*-*	Upregulated	[90]
	miR-16-5p	Human	Plasma	*ATG14 **	Upregulated	[46]
Apoptosis	miR-1, miR-30c, miR-149, miR-486	Mouse	Heart	-	Downregulated	[106]
	miR-29c	Mouse	Heart	*Camk1d **	Downregulated	[106]
	miR-30d	Mouse	Heart	*Ube2d3 **	Downregulated	[106]
	miR-499	Mouse	Heart	*Apc **	Downregulated	[106]
	miR-484	Mouse	PC, H9c2	LINC00339	Downregulated	[113]
	miR-675	Rat	Heart	*Pa2g4*	Upregulated	[112]
	miR-186-5p	Human	AC16	*XIAP*	Upregulated	[114]
	miR-23a-3p	Human	AC16	*CASP10*	Downregulated	[115]
	miR-150	Human	Plasma	*EGR2*	Downregulated	[111]
Fibrosis	miR-699a	Mouse	Heart	-	Downregulated	[130]
	miR-34a	Mouse	Heart	-	Upregulated	[133]
	miR-26, miR-30	Mouse/human	Plasma	-	Upregulated	[136]
	miR-208	Human	Heart	-	Upregulated	[129]
	miR-21	Human	Heart	-	Upregulated	[137]
	miR-29, miR-133	Human	Heart	-	Downregulated	[137]
	miR-33a	Human	Heart	-	Downregulated	[138]
	miR-221/-222	Human	Heart	-	Downregulated	[139]
	miR-125b	Human	Heart	-	Upregulated	[142]
	miR-144-3p, miR-9-3p	Human	Heart	*FN1 **	Downregulated	[143]
	miR-146a-5p	Human	Exosomes	-	Downregulated	[144]

## 9. Future Perspectives

The functional validation of miRNAs has enabled a better understanding of the cellular and developmental biology of various diseases at the molecular level [145]. As outlined in this review, miRNAs function as a regulator of the distinct mechanisms involved in DCM progression by targeting multiple signaling pathways. The ability to target multiple mRNAs that are altered in disease conditions makes these molecules remarkable candidates as therapeutics or as targets of therapeutics. The progression of miRNA-based drugs into preclinical and clinical testing of CVDs has evidenced this quality. Therefore, we speculated that a correction of the dysregulated miRNA might offer a potential strategy for the treatment of DCM. Nevertheless, more studies are still needed to better understand the significance of miRNAs in the different mechanisms involved in DCM in vivo and to delineate suitable strategies for developing therapies for DCM.

To date, clinical trials (phase 1 and 2) of antimiRs against miR-122-5p (miravirsen, RG-101) [146] have provided proof-of-principle evidence that miRNA-based therapy in patients is possible. Although the premature discontinuation of some trials reflects limitations that still need to be overcome, preclinical and clinical data obtained with inhibitory oligonucleotides provide valuable information for the design and performance of miRNA-targeting cardiovascular therapies. This applies to the miR-132-3p inhibitor (CDR132L) [147], which is currently planned for phase II testing and was developed to treat HF. Nonetheless, more complications have been observed in the application of miRNA mimics or overexpression for cardiovascular indications as ﻿timing and dosing seem particularly critical limitations [148].

On the other hand, numerous pharmacological challenges remain to be solved. Although the development of synthetic miRNA mimics and chemically modified antimiRs has helped us to overcome major hurdles, the delivery of these molecules, as well as the establishment of effective concentrations in target tissue, still pose considerable challenges. Their hydrophilic nature that hinders their membrane penetration, degradation in the bloodstream of systemically delivered oligonucleotides, as well as the poor delivery to the target site, in particular for cardiovascular tissue, of systemically delivered oligonucleotides are barriers that need to be overcome. Different approaches have been developed to circumvent these obstacles, of which many hold promise for cardiovascular applications, e.g., LNA antimiRs, as they penetrate membranes as “naked” molecules [149]. On the other hand, a large variety of strategies have been developed for the delivery of miRNA modulators to the cell type or tissue of interest, such as nanoparticles based on lipids, polymers, or metals, which serve as carriers of oligonucleotides [150], conjugation to targeting molecules (peptides, antibodies or other bioactive molecules), or encapsulation of miRNA mimics or the antimir into a lipid-based formulation that enhances cell-specific uptake. In this regard, cardiac-targeting peptides have proven their suitability in CVD models in vivo [151,152]. On the other hand, viral vectors represent smart vehicles for genetic information, including the adeno-associated virus vector BNP116, which is currently being tested in a phase I clinical study (ClinicalTrials.gov NCT04179643). However, this remains a largely underdeveloped area that needs to be deeply investigated.

Finally, further work is necessary to address some aspects that make data interpretation very challenging. When working with human samples, there can be conflicting observations regarding the miRNA expression profile in DCM. Since a large proportion of the current evidence is derived from monocentric case–control studies that, in most cases, lack external validation in large prospective cohorts, these differences can be attributed to potential differences in sample/patient numbers. In addition, co-morbidities should be considered. More importantly, there is a relevant lack of standardization in methods and analytical workflow, including sampling time, methods for miRNA quantification, as well as miRNA normalization parameters. Thus, the discovery of the ways in which one can avoid these limitations will be a major challenge in applying miRNAs for diagnostic and therapeutic purposes.

## Figures and Tables

**Figure 1 ijms-23-13573-f001:**
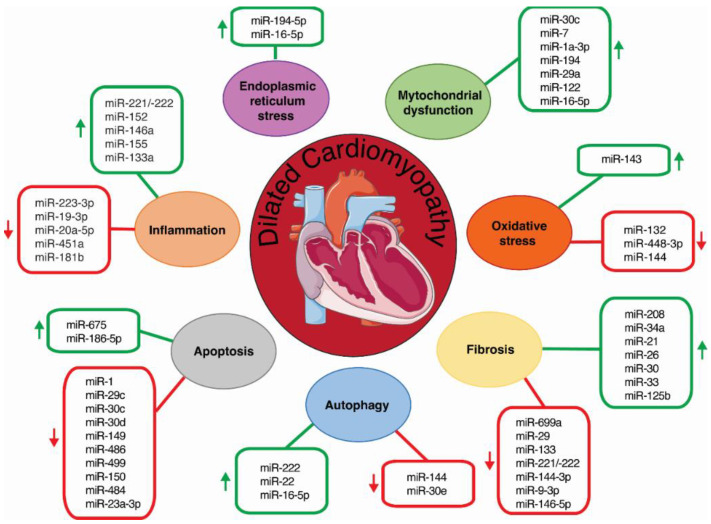
Schematic diagram depicts miRNAs orchestrating distinct mechanisms involved in DCM (inflammation, ER stress, mitochondrial dysfunction, oxidative stress, autophagy, apoptosis and fibrosis). Green and red arrows indicate positive and negative regulation in the disease, respectively.

**Figure 2 ijms-23-13573-f002:**
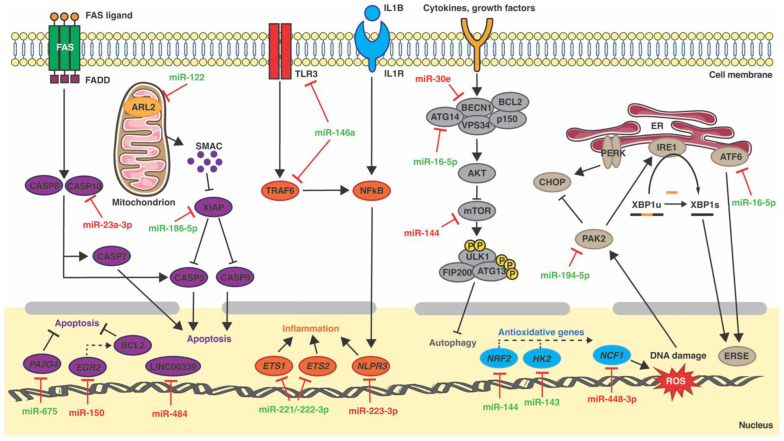
MiRNAs in cardiomyocyte inflammation, ER stress, mitochondrial dysfunction, oxidative stress, autophagy and apoptosis pathways. Green and red represent upregulated and downregulated miRNAs in the disease, respectively. FAS, fas cell surface death receptor; FADD, fas-associated death domain protein; CASP8, caspase-8; CASP7, caspase-7; CASP3, caspase-3; IL1R, interleukine-1 receptor; NFkB, nuclear factor kappa B; AKT, protein kinase B; mTOR, mechanistic target of rapamycin; CHOP, C/EBP homologous protein; ERSE, ER stress response element.

## Data Availability

In-data transparency is guaranteed. The datasets generated during and/or analyzed during the current study are available from the corresponding author on reasonable request.

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
