# Peer review of "The Role of MicroRNAs in Dilated Cardiomyopathy: New Insights for an Old Entity"

_ijms, 2022, doi:10.3390/ijms232113573_

Round 1

Reviewer 1 Report

THE ROLE MICRORNAS IN DILATED CARDIOMYOPATHY: NEW INSIGHTS FOR AN OLD ENTITY

Alonso-Villa E., et al.

The authors presented a well-written and updated review about how microRNAs are involved in the pathogenesis of dilated cardiomyopathy. The bibliography is accurate and specific to the subject, which is analyzed in detail. There are only a few minor issues with the use of English and some references that are not properly cited which I list below.

Line

76. Change “high” to “highly”

79. Delete “a” before “complementary sequences”. Otherwise, leave the “a” and change “sequences” to “sequence”.

196. Change “have shown” to “have been shown”.

314. “malondialdehyde” is in a different font from the rest of the manuscript.

329. “autophagy related 101” is in a different font from the rest of the manuscript. There should be a hyphen too “autophagy-related”.

400. Delete the “s” in “promotes”. It should say “promotes”.

401. “proliferation-associated 2G4” is in a different font from the rest of the manuscript.

510. Change “understand better” to “to better understand”.

837, 845, and 860. The reference only says “Pubmed”. The name of the journal, volume and pages should be added.

Reviewer 2 Report

The authors reviewed the recent progresses in the role of miRNAs in dilated cardiomyopathy. It is informative and constructive. My concerns are listed below:

1. I suggest the authors to provide more specific information in the abstract, including how miRNAs work. The current version of the abstract lacks the detailed progresses in the field.

2. Figure 1 showing the major miRNAs involved in dilated cardiomyopathy, it should be improved. What does the arrow stand for? How they affect the related processes, and how they contribute to the pathology of dilated cardiomyopathy? How they change in dilated cardiomyopathy? I suggest the authors to include the above information to make it more informative and accurate. 

3. I suggest the authors to provide extra figures to make the manuscript more readable and attractive.

4. It is well established that interventions such as exercise exert extensive cardioprotective effects against cardiac diseases, including dilated cardiomyopathy, and recent studies suggested that exercise exerts cardioprotection through regulation of miRNAs (PMIDs: 34606978, 33333247). I suggest the authors to include these progresses.

5. I suggest the authors to discuss the clinical relevance and progresses of the related progresses in the field in perspectives.

6. There is a big concern in the studies of miRNAs, controversy results and overstated conclusion were common. Please discuss the limitations and the concerns raised in these studies, which should be taken into consideration for data interpretation.

Reviewer 3 Report

What review stratergy is followed for shortlisting the papers ?

Mentions all the research questions addressed in this review?

Compare the review with existing reviews

Also, write the research gaps and potential research problems in this domain. 
